# Threat Model-Agnostic Adversarial Defense using Diffusion Models

## Abstract

Deep Neural Networks (DNNs) are highly sensitive to imperceptible malicious perturbations, known as adversarial attacks. Following the discovery of this vulnerability in real-world imaging and vision applications, the associated safety concerns have attracted vast research attention, and many defense techniques have been developed. Most of these defense methods rely on adversarial training (AT) – training the classification network on images perturbed according to a specific threat model, which defines the magnitude of the allowed modification. Although AT leads to promising results, training on a specific threat model fails to generalize to other types of perturbations. A different approach utilizes a preprocessing step to remove the adversarial perturbation from the attacked image. In this work, we follow the latter path and aim to develop a technique that leads to robust classifiers across various realizations of threat models. To this end, we harness the recent advances in stochastic generative modeling, and means to leverage these for sampling from conditional distributions. Our defense relies on an addition of Gaussian i.i.d noise to the attacked image, followed by a pretrained diffusion process – an architecture that performs a stochastic iterative process over a denoising network, yielding a high perceptual quality denoised outcome. The obtained robustness with this stochastic preprocessing step is validated through extensive experiments on the CIFAR-10 and CIFAR-10-C datasets, showing that our method outperforms the leading defense methods under various threat models.

## 1 Introduction

Deep neural network (DNN) image-classifiers are highly sensitive to malicious perturbations in which the input image is slightly modified so as to change the classification prediction to a wrong class. Amazingly, such attacks can be effective even with imperceptible changes to the input images. These perturbations are known as adversarial attacks Goodfellow et al. (2014); Kurakin et al. (2016); Szegedy et al. (2013). With the introduction of these DNN classifiers to real-world applications, such as autonomous driving, this vulnerability has attracted vast research attention, leading to the development of many attacks and robustification techniques.

Amongst the many types of adversarial attacks, the most common ones are norm-bounded to some radius $\epsilon$, where the norm $L_p$ and the radius $\epsilon$ define a threat model. The attack is posed as an optimization task in which one seeks the most effective deviation to the input image, $\delta$, in terms of modifying the classification output, while constraining this deviation to satisfy $\|\delta\|_p \leq \epsilon$. One way to robustify a network against such attacks is by training it to correctly-classify attacked examples from a specific threat model Madry et al. (2017); Zhang et al. (2019); Gowal et al. (2020). These methods, known as Adversarial Training (AT), lead to state-of-the-art performance when trained and tested on the same threat model. However, a well-known limitation of such methods is their poor generalization to unseen attacks, which is discussed in length in Hendrycks et al. (2021); Bai et al. (2021) as one of the unsolved problems of adversarial defense.

A different type of robustification techniques proposes a preprocessing step before feeding the image into the classifier Song et al. (2017); Samangouei et al. (2018); Yang et al. (2019); Grathwohl et al. (2019); Du & Mordatch (2019); Hill et al. (2020); Yoon et al. (2021). Since an adversarial example can be seen

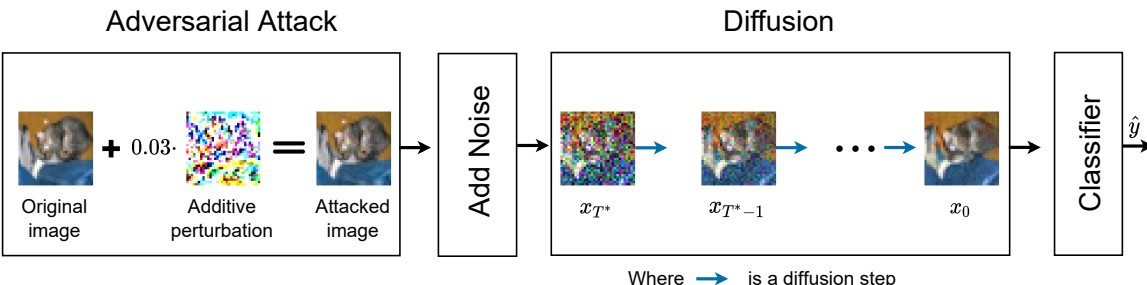

Figure 1: Our method flow. In the "Adversarial Attack" block, an attacker calculates the attack "Additive perturbation" and adds it to the "Original image" in order to create the "Attacked image". As a preparation for the diffusion process, in the "Add Noise" block, we add an i.i.d Gaussian noise to the attacked image according to Equation 3. We proceed by feeding it into the "Diffusion" block, consisting of diffusion steps that include a denoising and an addition of a Gaussian noise. This effectively samples a new image from the diffusion model initialized by $y$, the noisy attacked image (see more in Section 2.2). Lastly, we feed the preprocessed obtained image to a classifier.

as a summation of an image and an adversarial perturbation $\delta$, using such a procedure to remove or even attenuate this second term is reasonable. The authors of Song et al. (2017); Samangouei et al. (2018); Grathwohl et al. (2019); Du & Mordatch (2019); Hill et al. (2020); Yoon et al. (2021) use a generative model in the preprocessing phase in various ways. They either use the pretrained classifier directly or re-train a classifier on the generative model's outputs. In general, these kind of methods are very appealing since they are capable of robustifying any publicly-available non-robust classifier and do not require a computational expensive specialized adversarial training. Furthermore, such methods are oblivious of the threat model being used.

In this work we introduce a novel and highly effective preprocessing robustification method for image classifiers. We choose a preprocessing-based approach based on a generative model since we aim to remove or weaken the adversarial perturbation while effectively projecting it onto the learned image manifold, where the classifier's accuracy is likely to be high. While a generative model is typically used to sample from $p(x)$, the probability of images in general, our approach initializes this process with $y$ at the appropriate diffusion step, where $y$ is the noisy attacked image. This process effectively denoises the attacked image while targeting perfect perceptual quality Kawar et al. (2021b); Ohayon et al. (2021). More specifically, we use a diffusion model - an iterative process that uses a pretrained MMSE (Minimum Mean Squared Error) denoiser and Langevin dynamics. The later involves an injection of Gaussian noise, which helps to robustify our samplers against attacks, even if they are aware of our defense strategy. Our method relies on a preprocessing model and a classification one, where both are trained independently on clean images. Hence, our architecture is inherently threat model agnostic, achieving robustness for unseen attacks. In our experiments we propose a way to evaluate the threat model-agnostic robustness by presenting two measurements. The first is the average on a wide range of attacks, and the second is the average across the unseen attacks. We consider the following threat models: $(L_\infty, \epsilon = 8/255)$, $(L_\infty, \epsilon = 16/255)$, $(L_2, \epsilon = 1)$, $(L_2, \epsilon = 2)$. In summary, our main contributions are:

- A novel stochastic diffusion-based preprocessing robustification is proposed, aiming to be a model-agnostic adversarial defense.

- The effectivnes of the proposed defense strategy is demonstrated in extensive experiments, showing state-of-the-art results.

## 2 Background

### 2.1 Adversarial Robustness

Since the discovery of the phenomenon of adversarial examples in neural networks Goodfellow et al. (2014); Kurakin et al. (2016); Szegedy et al. (2013), classifiers' robustness has been extensively studied. Numerous works have been focusing on new methods for constructing adversarial examples and/or defending from them. In the following we bring the very fundamental results referring to adversarial defense and attack methods, as a background to our work.

Let us start with how adversarial attacks are created. Given an image $x$ and a classifier $f(\cdot)$, an adversarial attack is a small norm-bounded perturbation $\delta$, added to the input image $x$, that leads to its misclassification. There exist several mainstream settings for crafting adversarial examples that differ from each other in their assumptions regarding the defense method's characteristics and the access to the model and its gradients. We describe below such key attack configurations.

*White-Box Attacks* are applied when the attacker has full access to the full system architecture (including both the classifier and the defense mechanism), which is assumed to be differentiable. This is a rich and a widely used group of attacks that contains some of the most common ones, such as Fast Gradient Signed Method (FGSM) Goodfellow et al. (2014), Projected Gradient Decent (PGD) Madry et al. (2017) and CW Carlini & Wagner (2017). While there exist numerous white-box attack strategies, PGD is the cornerstone of their most modern embodiments. It is an iterative gradient-based algorithm that increases the classifier's loss in each step by perturbing the input data. We describe PGD in Algorithm 1 below.

---

**Algorithm 1** $L_\infty$-based Projected Gradient Descent

    **Input** classifier $f(\cdot)$, input $x$, target label $y$, norm radius $\epsilon$, step size $\alpha$, number of steps $N$
1: **procedure** PGD
2:     $\delta \leftarrow 0$
3:     **for** $i$ in $1 : N$ **do**
4:         $\delta \leftarrow \Pi_\epsilon(\delta + \alpha \cdot sign(\nabla_x Loss(f(x + \delta), y)))$
5:     **end for**
6: **end procedure**

---

The operator $\Pi_\epsilon$ is a projection onto the $L_p$ norm of radius $\epsilon$. In the $L_\infty$ case, $\Pi_\epsilon$ is just the clamp operation into $[-\epsilon, \epsilon]$.

Since white-box attacks have assumptions that do not always hold, they can not be used in every setup. For example, such a setup can be a defense method that relies on a non-differentiable preprocessing. Since white-box attacks are gradient-based, they are likely to fail in this case. Another example is stochastic preprocessing, which poses a challenging configuration for white-box attacks. This stems from the fact that the ideal crafted attack might not be optimal during inference due to randomness. In order to better adjust gradient-based adversarial attacks to such scenarios, alternative approaches were developed, as we describe hereafter.

*Grey-Box Attack* is used when the attacker has access to the classifier but not to the preprocessing model defending it, $g(\cdot)$. This approach is limited due to the fact that the attack in such a case is constructed upon $f(\cdot)$ while being evaluated with $f(g(\cdot))$. As a consequence, the malicious perturbation created is necessarily sub-optimal and thus less effective.

*Backward Pass Differentiable Approximation (BPDA) Attack* Athalye et al. (2018) is an attack method for cases in which the preprocessing function $g(\cdot)$ is non-differentiable or impractical to differentiate, implying that $f(g(\cdot))$ is not differentiable as well. In many cases we can invoke the assumption that $g(x) \approx x$, reflecting the fact that preprocessing methods do not perform significant modifications to the input images, but rather try to remove the already small malicious perturbations. In order to attack such architecture we use the forward pass of the preprocessing $g(\cdot)$ and approximate its derivative with $I$, producing $\nabla_x f(g(x)) \approx$

$\nabla_{g(x)} f(g(x))$. With this in place, the attacker can perform white-box attacks without completely disregarding the preprocessing steps.

*Expectation-Over-Transformation (EOT) Attack* Athalye et al. (2018) is used when the preprocessing step $g(\cdot)$ is stochastic. Attacking such a method is harder for gradient-based methods, since the crafted deviation vector $\delta$ might not remain optimal during inference due to the randomness. EOT calculates the attack's gradients by $\nabla_x \mathbb{E}[f(g(x))] = \mathbb{E}[\nabla_x f(g(x))]$, differentiating through both the classifier and preprocessing with an expectation. In practice, EOT empirically approximates the expectation with a fixed number of drawn samples from $g(x)$.

We move now to discuss adversarial defense approaches. In the past few years, numerous such methods were proposed to improve the robustness of classifiers to adversarial attacks. While there are many types of robustification algorithms, we focus below on two such families.

*Adversarial Training (AT) Defense* proposes to utilize adversarial examples during the training process of the classifier. More specifically, the idea is to train the model to classify such examples correctly. Several recent works Madry et al. (2017); Zhang et al. (2019); Gowal et al. (2020) follow this line of reasoning, leading to the current state-of-the-art in robustifying classifiers.

*Preprocessing* is a substantially different type of robustification method that relies on a preceding operation on the classifier's input as its name suggests. Since adversarial examples contain small imperceptible perturbations, using preprocessing steps to "clean" them seems to be an is intuitive step. Many works rely on various generative models for such preprocessing Song et al. (2017); Samangouei et al. (2018); Du & Mordatch (2019); Hill et al. (2020); Yoon et al. (2021). More specifically, these models are used to project the attacked image into a valid clean one in its vicinity, with the hope that the processed image is more likely to be classified correctly.

## 2.2 Diffusion Models

Diffusion models Sohl-Dickstein et al. (2015); Ho et al. (2020); Song & Ermon (2019) are Markov Chain Monte Carlo (MCMC)-based generative techniques, which consist of a chain of images $x_0, x_1, ..., x_T$ of the same size as the given image $x$. These methods are based on two closely related processes. The first is the forward process of gradually adding Gaussian noise to the data according to a decaying variance schedule parametrized by $1 > \alpha_0 > \alpha_1 > \cdots > \alpha_T > 0$. The following defines this chain of steps, for $t = 1, 2, \ldots, T$ where $x_0$ is the given clean image $x$:

$$q(x_t|x_{t-1}) := \mathcal{N}\left(\sqrt{\frac{\alpha_t}{\alpha_{t-1}}}x_{t-1}, \left(1 - \frac{\alpha_t}{\alpha_{t-1}}\right)I\right) \tag{1}$$

Posed differently, the forward process can be described as a simple weighting between the image $x_0$ and a Gaussian noise vector,

$$q(x_t|x_0) = \mathcal{N}(\sqrt{\alpha_t}x_0, (1 - \alpha_t)I), \tag{2}$$

so we can express $x_t$ as

$$x_t = \sqrt{\alpha_t}x_0 + \sqrt{1 - \alpha_t}\epsilon; \quad \epsilon \sim \mathcal{N}(0, I). \tag{3}$$

When $\alpha_t$ is close to zero, $x_t$ is close to a pure standard Gaussian noise, independent of $x_0$. Thus, we can set $x_T \sim \mathcal{N}(0, I)$ as initialization for the backward process, which is explained next.

The second and the more intricate process is the backward direction, which gradually removes the noise from the image. Intuitively, this stage denoises the image by pealing layers of noise gradually. A key ingredient in this process is a pretrained noise estimator neural network, $\epsilon_\theta(x_t, t)$. This denoiser serves as an approximation to the score function $\nabla \log p(x)$ Kadkhodaie & Simoncelli (2020), bringing the knowledge about the image

**Diffusion**

Figure 2: Our method incorporates a diffusion model and a classifier. In every diffusion step, we add Gaussian noise multiplied by the corresponding $\sigma_t$, which is a user-controlled hyperparameter. The variables $x_{T^*}, ..., x_1$ constitute the MCMC, and the last step's output of the diffusion model $x_0$, is the final output, to be sent to the classifier.

statistics into this sampling procedure. The noise estimator is conditioned on the time $t$, trying to estimate the noise $\epsilon$ of the latent variable $x_t$. Sampling, or generating an image, is performed by iteratively applying the following update rule for $t = T, T - 1, \ldots, 0$:

$$x_{t-1} = \sqrt{\alpha_{t-1}}\left(\frac{x_t - \sqrt{1 - \alpha_t}\epsilon_\theta(x_t, t)}{\sqrt{\alpha_t}}\right) + \sqrt{1 - \alpha_{t-1} - \sigma_t^2}\epsilon_\theta(x_t, t) + \sigma_t\epsilon_t \tag{4}$$

where the first term is a denoising stage – an estimation of $x_0$, while the second term stands for an attenuated version of the estimated additive noise in $x_t$. $\sigma_t\epsilon_t$ is a stochastic addition, where $\sigma_t$ is a hyperparameter controlling the stochasticity of the process, and $\epsilon_t \sim \mathcal{N}(0, I)$.

The sampling process posed in Equation (4) tends to be very slow, requiring $T$ ($\approx 1000$) passes through the denoising network. Methods for speeding up this process are discussed in Nichol & Dhariwal (2021); Song et al. (2020); Kawar et al. (2022). There are various use-cases for diffusion models beyond image synthesis. The ones relevant to our work are discussed in Meng et al. (2021); Kawar et al. (2021b;a; 2022) where inverse problems are being considered. Following Meng et al. (2021), instead of sampling from the ideal image distribution $p(x)$, the diffusion process we implement is initialized with $x_{T^*}$, where $x_{T^*}$[1] is the given noisy image. Thus, the outcome $x_0$ can be considered as a stochastic high perceptual quality denoising of $x_{T^*}$.

## 3 Our Method

In this section we present our adversarial defense method, depicted in Figure 1. We start by adding noise to the attacked image, and then proceed by preprocessing the obtained image using a generative diffusion model, effectively projecting it onto the learned image manifold. The outcome of this diffusion is fed into a vanilla classifier, which is trained on the same image distribution that the diffusion model attempts to sample from. Thus, our framework is comprised of two main components – a denoiser that drives the diffusion model and a classifier.

Intuitively, we would like to sample images that are semantically close to an input image $x$ by starting the diffusion process from some intermediate time step ($T^* < T$) rather than the beginning ($T^* = T$). Recall that $x_T$ stands for a pure Gaussian noise, whereas $x_{T^*}$ would be the noisy image we embark from. To this end, we modify the image to fit the diffusion model at this time step by applying Equation 3 – simply multiplying $x$ by a scalar and adding an appropriate Gaussian noise, resulting in $x_{T^*}$. We feed this processed image into the diffusion model at time step $T^*$ and complete the diffusion process, running with $t = T^*, T^* - 1, \ldots, 0$, and outputting $x_0$. Such a partial diffusion is similar to the image editing process presented in Meng et al. (2021), and close in spirit to the posterior sampler that is discussed in Kadkhodaie & Simoncelli (2020). We provide a comprehensive description of our method in Algorithm 2.

---

[1]More on the relation between $T$ and $T^*$ is given below.

---

**Algorithm 2** Our Preprocessing Defense Method

---

**Input** image $x$, maximum depth $T^*$, diffusion model denoiser $\epsilon_\theta(\cdot, \cdot)$,
variance schedule $[\alpha_T, \ldots, \alpha_0]$, stochasticity hyperparameters $[\sigma_T, \ldots, \sigma_1]$,

1: **procedure** SAMPLING
2:      $\epsilon_{T^*} \sim \mathcal{N}(0, I)$
3:      $x_{T^*} \leftarrow \sqrt{\alpha_{T^*}} x + \sqrt{1 - \alpha_{T^*}} \epsilon_{T^*}$
4:      **for** $t$ **in** $[T^*, T^* - 1, ..., 1]$ **do**
5:          $\tilde{x}_{t-1} \leftarrow \frac{x_t - \sqrt{1 - \alpha_t} \epsilon_\theta(x_t, t)}{\sqrt{\alpha_t}}$
6:          $\epsilon_t \sim \mathcal{N}(0, I)$
7:          $x_{t-1} \leftarrow \sqrt{\alpha_{t-1}} \tilde{x}_{t-1} + \sqrt{1 - \alpha_{t-1} - \sigma_t^2} \epsilon_\theta(x_t, t) + \sigma_t \epsilon_t$
8:      **end for**
9:      **return** $x_0$
10: **end procedure**

---

An important hyperparameter for the success of our method is the initial diffusion depth $T^*$, since different values of it yield significant changes in $x_0$. To better understand the importance of a careful choice of $T^*$, we intuitively analyze its effect. On the one hand, when starting from $T^* = T$, we sample a random image from the generative diffusion model, which obviously eliminates the adversarial perturbation. However, as the resulting image is independent of $x$, this will necessarily change class-related semantics of the image, which in turn would lead to misclassification. On the other hand, choosing $T^* = 0$ results in the same input image $x$, which does not remove the perturbation from the image, hence probably leading to misclassification as well. In other words, we need to choose $T^*$ that balances the trade-off between cleaning the adversarial noise, and keeping the semantic properties of the input image $x$. Choosing such $T^*$ that successfully balances these properties is crucial to the success of our adversarial defense algorithm.

We utilize the above described sampling algorithm with one goal in mind – sampling an image that is not contaminated with an adversarial attack while keeping it semantically similar to the original input image $x$. We believe that our algorithm is suited for this task because the Gaussian noise injections are much larger than the adversarial perturbation. Hence, the noise overshadows the adversarial attack, reducing its effect. This leads to a sampling process that answers both of our demands, removal of the contamination while remaining semantically close to $x$.

As mentioned previously, our method is comprised of a diffusion model denoiser and a classifier, both trained on clean images. This framework is very useful from a practical point of view, since we can utilize publicly available pretrained models to a completely different task than they were trained on – adversarial defense. The fact that these models were trained without adversarial attacks in mind gives our method a significant advantage – it is inherently threat model-agnostic. This essentially avoids the challenged generalization to unseen attacks problem Hendrycks et al. (2021); Bai et al. (2021), according to which classifiers trained on a specific adversarial threat model are vulnerable to attacks under a different threat regime.

A method close in spirit to ours is the Adaptive Denoising Purification (ADP) Yoon et al. (2021), which uses a score-based model as an adversarial defense. Despite this similarity, there are some fundamental differences that we would like to highlight. ADP suggests a score-based gradient ascent algorithm as a preprocessing step for robustifying a pretrained classifier. More specifically, they add Gaussian noise to the input image only at the beginning, and then apply a *deterministic* gradient ascent process with an adaptive step size. In contrast, we propose a stochastic diffusion-based preprocessing step, in which we inject noise into every diffusion iteration. This effectively samples from the learned image distribution, initialized with a noisy version of the input image. The increased stochasticity is a key property of our method that enables us to wipe the malicious attack, while effectively projecting the attacked image onto the learned image manifold, achieving robustness to unseen attacks.

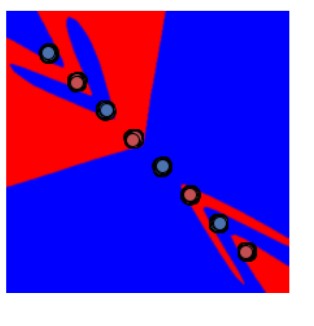
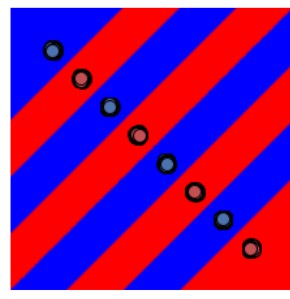

(a) Original classifier            (b) Our method

Figure 3: Decision boundary comparison between a vanilla classifier with and without our method on a 2D synthetic dataset.

# 4 Experiments

We proceed by empirically demonstrating the improved performance attained by our proposed adversarial defense method. First, we provide supporting evidence for our method when applied to a synthetic dataset. Next, we compare our method with another preprocessing method Yoon et al. (2021) under grey-box, BPDA+EOT, and white-box attacks. Following, we compare our method to various state-of-the-art (SoTA) methods on white box attacks, and finally present results over CIFAR-10-C. Additional experiments are reported in the supplementary material.

Throughout our experiments, we use the pretrained diffusion model from Song et al. (2020) and a vanilla classifier, both trained on clean images from CIFAR-10 Krizhevsky (2009) train set (50,000 examples). More specifically, we set the diffusion model maximal depth to $T^* = 140$ and the sub-sequence of the time steps to $\tau = \{T^*, T^* - 10, \dots, 10, 0\}$. In addition, we use a WideResNet-28-10 Zagoruyko & Komodakis (2016) architecture as our classifier and evaluate the performance on the CIFAR-10 test set (10,000 examples).

## 4.1 Synthetic Dataset Experimets

We create a synthetic 2D dataset (see Figure 3) and investigate the effect of a diffusion process on the decision boundaries of the classification. The dataset consists of two classes – red and blue points – consisting altogether of $10,000$ examples, drawn from two mixtures of Gaussians, each consisting of 4 concentrated groups. We train a fully connected neural network model to classify this data, having 10 layers of width 128. The training is done via $5,000$ epochs. As for the diffusion preprocess, we use an analytic score-function $\nabla \log p(x)$ of the known distribution, following the work of Song & Ermon (2019). We set $T^* = 10$ and values of $\alpha$ in the range $[0.1, 1]$.

After training the classifier, we calculate its decision rule and present it in Figure 3a, where the background colors represent the predicted label. As can be seen, the classifier achieves perfect performance, as all the red points are located in the red zone, and all the blue ones are surounded by a blue background. Nevertheless, the classifier decision boundaries are very close to the data, which is a well-known phenomenon of vanilla classifiers Shamir et al. (2021). This illustrates why small perturbations to the data, such as adversarial attacks, can change the classification decision from the correct to the wrong ones.

When applying our preprocessing scheme, our method leads to a larger margin between the data points and the decision boundaries, as can be seen in Figure 3b. These results are encouraging because in the adversarial attack regime, every data point is allowed to perturbed with an $\epsilon$ norm ball around it. When the decision boundaries are far enough from the data points, an $\epsilon$-bounded attack would necessarily fail.

Table 1: CIFAR-10 robust accuracies of preprocessing methods under the following attack: grey-box, BPDA + EOT, white-box PGD. All using the same threat model $L_\infty, \epsilon = 8/255$.

| Defense | Attack | Base Classifier | | Preprocessed | |
|---------|--------|------|------|------|------|
| | | Clean | Adversarial | Clean | Adversarial |
| ADP Yoon et al. (2021) | grey-box | 95.60 | 00.00 | 86.39 | 80.49 |
| **Ours** | grey-box | 95.60 | 00.00 | 86.28 | **82.33** |
| ADP Yoon et al. (2021) | BPDA+EOT | 95.60 | 00.00 | 86.39 | 44.79 |
| **Ours** | BPDA+EOT | 95.60 | 00.00 | 86.28 | **77.65** |
| ADP Yoon et al. (2021) | white-box | 95.60 | 00.00 | 86.39 | 31.42 |
| **Ours** | white-box | 95.60 | 00.00 | 86.28 | **63.40** |

Table 2: CIFAR-10 robust accuracies under white + EOT attacks. For every compared method, we state the threat model that was used in training in the first column Trained Threat Model (TTM) column. The next four columns are the four different threat models used for evaluation. The next two columns are the two averages that we use for evaluation, Average without Training (AwT), and Average of All (AoA). In the last column we state the classifier architecture that is used.

| Method | TTM | Attack | | | | AwT | AoA | Architecture |
|--------|-----|--------|--------|-----|-----|-----|-----|--------------|
| | | $L_\infty$ | | $L_2$ | | | | |
| | | 8/255 | 16/255 | 1 | 2 | | | |
| AT Madry et al. (2017) | $L_\infty, \epsilon = 8/255$ | 54.23 | 19.20 | 32.34 | 04.99 | 18.84 | 27.69 | rn-50 |
| | $L_2, \epsilon = 0.5$ | 34.25 | 02.99 | 41.55 | 05.72 | 21.13 | 21.13 | rn-50 |
| Trades Zhang et al. (2019) | $L_\infty, \epsilon = 8/255$ | 55.79 | 23.18 | 32.51 | 05.01 | 20.23 | 29.12 | wrn-34-10 |
| Gowal et al. Gowal et al. (2020) | $L_\infty, \epsilon = 8/255$ | 66.35 | 34.81 | 41.87 | 09.62 | 28.77 | 38.16 | wrn-28-10 |
| | $L_2, \epsilon = 0.5$ | 47.08 | 13.12 | 52.71 | 14.85 | 31.94 | 31.94 | wrn-70-16 |
| PAT - Laidlaw et al. (2020) | | 44.07 | 22.33 | 46.65 | 23.33 | 34.01 | 34.01 | rn-50 |
| Ours | | 51.05 | 37.76 | 50.75 | 19.23 | **39.70** | **39.70** | wrn-28-10 |

## 4.2 CIFAR-10 Experimets

First, we compare our method to ADP Yoon et al. (2021), a leading preprocessing method, using the following attacks: grey-box, BPDA+EOT, and white-box, where the EOT is approximated over 20 repetitions. As can be seen in Table 1, our method outperforms ADP by up to 32.86%. We should note that the results are lower than presented in Yoon et al. (2021), this was also observed in Croce et al. (2022).

Next, we compare our method to baseline state-of-the-art (SoTA) methods, under PGD attacks using four different threat models – $(L_2, \epsilon = 1)$, $(L_2, \epsilon = 2)$, $(L_\infty, \epsilon = 8/255)$, $(L_\infty, \epsilon = 16/255)$- more details are given in supplementary material. To assess the generalization ability to unseen attacks, we average the results in two ways: (i) *Average of All*: accuracy average of all the attacks; and (ii) *Average of Unseen Attack*: accuracy average of the attacks not seen at training time (if applicable). While the first is a simple average that also considers the performance on the attack used in training time, the second showcases the generalization capabilities to unseen attacks. Note that because our method is not trained on any threat model, (i) and (ii) are the same. As can be seen in Table 2, adversarial training methods excel on the specific threat model that they trained on. However, they generalize poorly, as discussed in Bai et al. (2021); Hendrycks et al. (2021), while our method achieves SoTA performance in both of the examined metrics.

### 4.3 Robustness to CIFAR-10-C Perturbations

In most of our discussion we focused on a robustness to norm- bounded attacks. We turn now to introduce a robust classification under attacks that are based on augmentation. These refer to modifications of the image in various ways such as motion blur, zoom blur, snow, JPEG compression, contrast variation, etc. CIFAR-10-C Hendrycks & Dietterich (2019) is such a corrupted images dataset that was created by performing numerous augmentations on CIFAR-10 Krizhevsky (2009) dataset. CIFAR-10-C is commonly used for evaluating the robustness performance under broad attacks.

As our method is inherently attack agnostic, it is natural to evaluate it on this class of attacks. We compare our method versus other leading techniques, achieving state-of-the-art results. This experiment requires adjustment of the diffusion model maximal depth parameter $T^*$. When we set $T^* \in [30, 90]$, we outperform the other methods, as depicted in Figure 4.

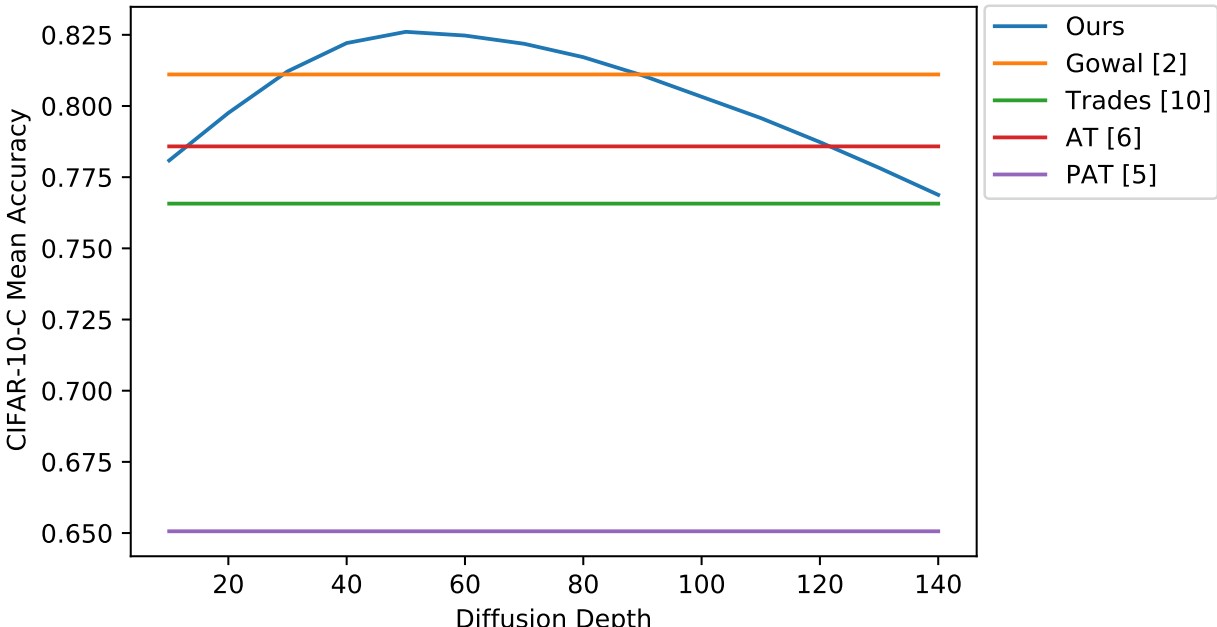

Figure 4: Robustness accuracy under CIFAR-10-C as a function of the diffusion model maximal depth $T^*$. We compare our method with the results reported in Gowal et al. (2020); Zhang et al. (2019); Madry et al. (2017); Laidlaw et al. (2020).

### 4.4 Diffusion Depth and Sampling

When deploying the proposed diffusion defense, two critical parameters should be discussed - the choice of $T^*$ (referred to as depth) and the time-step skips to use. In this Subsection we discuss the effect of both.

We start by showing the influence of the depth of the diffusion model on the robust accuracy. As we change the maximal depth of the diffusion model $T^*$, we depict the robust accuracy obtained by our method, and present it in Figure 5. As discussed in Section 3, the diffusion depth controls the trade-off between clearing the attack perturbation and sampling an image that is semantically similar to the input image $x$. We track the diffusion model behavior as we increase the diffusion model's first step. When setting $T^*$ to a shallow diffusion step, we effectively sample images that are closer to the input image $x$, and since the image is contaminated by a malicious attack, the classification accuracy is low. As we increase the depth we reach a sweet-spot in which we clean the malicious perturbation while keeping a small perceptual distance to $x$, which leads to the highest accuracy. When the depth is too big, we clear the attack but lose perceptual similarity to $x$, and the accuracy is reaching 10%, meaning that we sample random images.

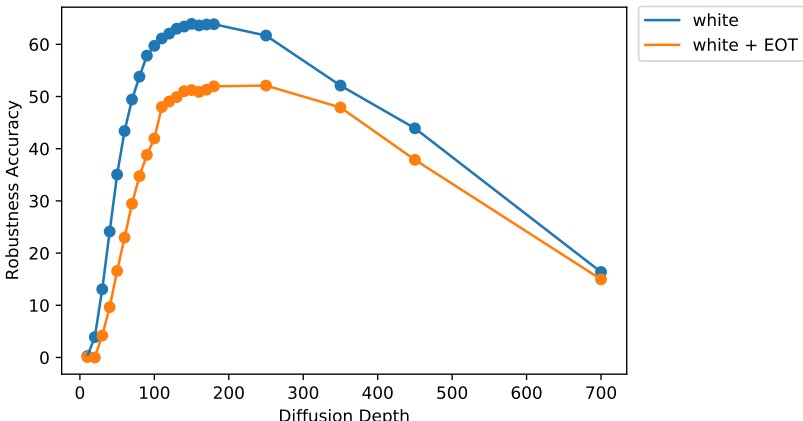

Figure 5: The obtained robust accuracy under white box attacks as a function of the max depth $T^*$ of the diffusion model. There are two graphs, both are attaked using the same threat model $L_\infty, \epsilon = 8/255$, the first is the robust accuracy under white-box attack, and the other refers to a white-box + EOT.

We now move to explore the influence of the skips to the time-steps in the diffusion process. Attacking our preprocessing method necessarily consumes a lot of time and memory, making it hard to break, as indeed claimed in Hill et al. (2020). This is due to the fact that an attack process requires keeping a computational graph of all the time steps of the diffusion process for computing derivatives. In contrast, our defense mechanism is lighter, as no derivatives are required, and only $T^*$ forward passes through the denoiser are performed.

When evaluating our defense method under the strongest known attack, white-box + EOT, we must lighten further our protection by reducing the number of diffusion steps. This is done by using only 1/10 of the DDIM diffusion steps Song et al. (2020), requiring all-together 14 steps. For uniformity of our experiments, we use this sub-sequence of steps for all attacks.

We should note that if the proposed preprocess diffusion is applied in full (no subsampling), this would increase both the attack and defense runtime and memory consumptions by a factor of 10. Such an approach would not worsen the robust accuracy, and perhaps even improve it, as can be seen in the supplementary material. Both these effects have one clear conclusion – when using our defense in practice, we can increase the diffusion model sampling, harming the attacker, while preserving the robust accuracy.

## 5   Related Work

The goal of preprocessing methods is to clean the adversarial attacks from the input images, leading to correct prediction by deep neural network classifier. Preliminary work on preprocessing defense methods include rescaling Xie et al. (2017), thermometer encoding Buckman et al. (2018), feature squeezing Xu et al. (2017), GAN for reconstruction Samangouei et al. (2018), ensemble of transformations Raff et al. (2019), addition of Gaussian noise Cohen et al. (2019) and mask and reconstruction Yang et al. (2019). It was shown by Athalye et al. (2018); Tramer et al. (2020) that such preprocessing, even if it includes stochasticity and non-diferentiability, can be broken when evaluated properly by adjusting the projected-gradient-descent attack, using backward-pass-differentiable-approximation and expectation-over-transformation algorithms. A new preprocessing group of work has recently emerged, trying to utilize Energy-Based-Model (EBM) to the task of cleaning adversarial pertubation from images. The intuition is that generative models are capable of sampling images from the image manifold, hopefully projecting attacked images that were deviated from the image manifold, back onto it. To this end, some EBM preprocessing methods were developed: purification by pixelCNN Song et al. (2017), restore corrupt image with EBM Du & Mordatch (2019) and density aware classifier Grathwohl et al. (2019). Most recent methods includes: long-run Langevin sampling Hill et al. (2020) and gradient ascent score based-model Yoon et al. (2021). In contrast to many of these methods

that require retraining the classifier, our method does not have this requirement, the diffusion model and classifier are both pretrained on clean images.

Defense to unseen attacks methods: Recently, an attention for defense to unseen attacks has emerged. Previouse methods that include Adversarial Training (AT) do not generalize well to unseen attacks, as shown in Hendrycks et al. (2021); Bai et al. (2021). For this end, a new robustness evaluation metric to unseen attacks was suggested Kang et al. (2019). Moreover, the authors of Laidlaw et al. (2020) suggested perceptual-adversarial-training, which takes into account the perceptual similarity, leading to a new method that generalizes to unseen attacks.

## 6 Conclusion

This work presents a novel preprocessing defense mechanism against adversarial attacks, based on a generative diffusion model. Since this generative model relies on pretraining on clean images, it has the capability to generalize to unseen attacks. We evaluate our method across different attacks and demonstrate its superior performance. Our method can be used to defend against any attack, and does not require retraining the vanilla classifier.

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
