# OpenReview forum: "Threat Model-Agnostic Adversarial Defense using Diffusion Models"
_TMLR — Withdrawn by Authors_

### Review · Reviewer_Eawt · 2022-12-12

**Summary Of Contributions:**

This paper proposes a diffusion model-based denoising approach for adversarial defense. The idea is to purify an adversarially perturbed image by first adding large gaussian noise to the image and then applying the diffusion model to recover the image while removing the adversarial effect of the image. Both the diffusion model and the classifier are trained on clean images. The key is to leverage a large amount of noise that can be handled by the diffusion model to overwrite the small adversarial noise. Experiments proved the advantage of this approach: robust to different threat models and attacks (even common corruptions).

**Audience:**

Yes

**Broader Impact Concerns:**

No ethical concerns are identified.

**Claims And Evidence:**

No

**Requested Changes:**

1. Evaluation against white EOT with at least 1k repetitions.

2. Results on ImageNet.

3. Evaluation against strong adaptive attacks and AutoAttack.

4. Exploration of adversarially trained diffusion models.

5. Exploration of the combination with AT method (for the classifier).

6. Exploration of whether diffusion models are naturally more robust to adversarial noise.


**Strengths And Weaknesses:**

Pros:
1. An interesting application of diffusion models for adversarial defense.

2. The proposed input denoising defense is independent of AT methods.

3. The denoising is threat model agnostic.


Cons:
1. Missing evaluation of adaptive attacks. For instance, an adaptive attack could attack the diffusion model to output an arbitrary image that is far away from x.

2. The white+EOT evaluation is quite weak, should test repetitions up to ~1k - 10k (only 20 was examined in the paper) to see the robustness drop, as diffusion models are noise/randomness heavy models.

3. Limited technical novelty. The proposed framework is a straightforward application of diffusion models for denoising, not really linked to adversarial defense/training.

4. Only tested on low reosultion cifar-10 dataset.

5. Not evaluated against the AutoAttack.

---

### Review · Reviewer_k9RQ · 2022-12-17

**Summary Of Contributions:**

This paper proposes a novel stochastic diffusion-based preprocessing defense method to improve adversarial robustness for image classifiers. The proposed defense is model-agnostic, can be used to defend against any attack, and does not require retraining the vanilla classifier. It conducts extensive experiments to show the effectiveness of the proposed defense strategy.

**Audience:**

Yes

**Broader Impact Concerns:**

I don't have any concerns about the ethical implications of the work.

**Claims And Evidence:**

No

**Requested Changes:**

1. It should include enough details about the attacks used. It should evaluate the proposed defense under the attacks proposed in [1] (e.g. AutoPGD, RayS, and Transfer attacks). This is very critical for acceptance.

2. It should ensure fair comparisons with the baselines. It should explain why the comparisons are fair. This is very critical for acceptance.

3. It should describe how to tune the hyper-parameter $T^*$ in the proposed defense. This is very critical for acceptance.

4. The claims made in the paper should be well-supported by the experimental results. This is very critical for acceptance.

**Strengths And Weaknesses:**

I think this paper has the following strengths:

1. It is well-written and easy to understand. It includes enough details for the proposed method.

2. The related works are appropriately discussed.

3. The proposed defense method is novel and achieves good results under the attack evaluation.

However, I think this paper has the following weaknesses:

1. The biggest concern is that the attacks considered may not be strong enough and it may overclaim the adversarial robustness achieved. It doesn't include enough details about the attacks (e.g., how many attack steps are used). It doesn't describe how the white-box attack is performed for the proposed method (e.g., what is the attack objective). I think it should also evaluate the proposed defense under the attacks proposed in [1] (e.g. AutoPGD, RayS, and Transfer attacks).

2. When comparing the proposed method with the baselines, it needs to ensure that the comparisons are fair (e.g. make sure that they are evaluated under similar attacks with a similar attack strength). Since it doesn't describe the attacks used for the proposed defense method and the baselines in detail, it is hard for me to evaluate if the comparisons are fair or not. In Table 2, different methods use different kinds of model architectures. I think for fair comparisons, it needs to make the architectures to be the same for all the methods.

3. It mentions that an important hyper-parameter for the success of the proposed defense method is the initial diffusion depth $T^*$. It needs to choose $T^*$ that can balance the trade-off between cleaning the adversarial noise and keeping the semantic properties of the input image $x$. I am wondering if it is easy to select a good $T^*$ in practice. I think it needs to provide a method to select the hyper-parameter $T^*$. Note that it should not tune $T^*$ using the test data. Figure 4 shows that when $T^* \in [30, 90]$, the proposed method outperforms the other methods. It needs to explain how such $T^*$ is tuned without using the test data.

4. In the conclusion, it claims that the proposed method can be used to defend against any attack. I think this claim is not well-supported by the experiments. In the experiments, it only evaluates the proposed method under the $L_p$ norm attacks and the image corruptions. There are some other types of attacks. If it wants to claim the defense can work for any attack, then it needs to add more experiments on other kinds of attacks.


[1] Croce, Francesco, et al. "Evaluating the Adversarial Robustness of Adaptive Test-time Defenses." arXiv preprint arXiv:2202.13711 (2022).

---

### Review · Reviewer_ve72 · 2022-12-19

**Summary Of Contributions:**

This paper proposes to use diffusion models to preprocess the image by injecting Gaussian noise, denoising with diffusion models, and feeding into a pre-trained image classifier.

Experiments on CIFAR-10 and CIFAR-10-C demonstrate that this strategy achieves superior robustness in terms of average robust accuracy among both L2/Linf norms and multiple perturbation magnitudes against whitebox/graybox + EOT attacks, and superior robustness on CIFAR-10-C, compared to adversarial training based training approaches and Adaptive Denoising Purification.

**Audience:**

No

**Broader Impact Concerns:**

A broader impact statement may need to be added to discuss how the defense should be viewed. Especially, the last sentence "Our method can be used to defend against any attack, and does not require retraining the vanilla classifier." is a bit overclaiming, since "any attack" should be refined to "commonly-known input evasion attacks that incur small perturbations".

**Claims And Evidence:**

Yes

**Requested Changes:**

- Experiments are a bit limited - more experiments on CelabA or ImageNet would significantly strengthen the paper.
- Clearly state the difference and conduct possible experimental comparisons with [1].

Minor:
1. Figure 5, caption: attaked -> attacked
2. Figure 4, the accompanying number brackets for compared baselines Gowal, Trades, AT, and PAT are meaningless. Since the reference list has no index number.

**Strengths And Weaknesses:**

Strengths:
- An effective adversarial robust defense approach based on a simple but effective strategy, leveraging the recent breakthroughs in diffusion models.

- Experiments on CIFAR-10 and CIFAR-10-C highlight the threat model-agnostic ability.

Weaknesses:
- The method, experimental evaluation, and main findings share much overlap with [1]. According to my understanding, the empirical performance and theoretical analysis of [1] seems to be more comprehensive and insightful. However, the submission does not cite nor compare with [1].


[1] Nie, W., Guo, B., Huang, Y., Xiao, C., Vahdat, A., & Anandkumar, A. (2022). Diffusion Models for Adversarial Purification. ICML 2022.

---

### Review · Reviewer_26ts · 2022-12-19

**Summary Of Contributions:**

This paper proposes a new preprocessing-based defense against adversarial examples. The authors utilize a diffusion model as a denoising mechanism that counteracts the adversarial perturbation. The defense starts by adding Gaussian noise to an adversarially-attacked input. Mingling adversarial noise with Gaussian noise, the authors dilute the effect of adversarial noise. Then, as a diffusion model denoises the Gaussian noise, a benign input is recovered while preserving the weakened adversarial effect. Because the preprocessing does not assume anything about the attack type, this defense method is threat model-agnostic.
The paper contains four different experiments. First, to show the effect of the preprocessing method, the authors visually present the change of decision boundaries before and after the preprocessing. Second, the authors evaluate the performance of the proposed defense under various attack scenarios, with comparisons to existing methods: preprocessing-based defense and adversarial training. Third, the authors tested the robustness against other types of perturbations, such as blurring and contrast variation. Finally, the authors empirically explore the effect of diffusion depth choice.

**Audience:**

Yes

**Broader Impact Concerns:**

I don’t see a particular broader impact concern regarding this paper.

**Claims And Evidence:**

Yes

**Requested Changes:**

As pointed out as a weakness, this paper needs more empirical support for the method. To improve the paper’s quality, I suggest the following experiments.

1. The synthetic data experiments should show more information. Currently, this experiment only shows a diffusion process's “before and after” on a single synthetic dataset.
    - The authors want to show the effect of a diffusion process. Because a diffusion process is a “process,” people may want to see how the decision boundary changes over time. Thus, it would be a better idea to show the intermediate output of the method: the decision boundary computed from the intermediate outputs of the process.
    - Does a diffusion-based defense provide a better alignment than other preprocessing-based defenses? If so, the authors should present the comparison because it will put more value on the proposed method.
    - Also, the authors may use synthetic data to show the effect of diffusion depth visually.
    - Even for experiments with synthetic data, one dataset looks like a small number of datasets, and readers may consider this insufficient support.
2. The authors should make the experiments more comprehensive. The authors used various attack strategies and adversarial training methods; however, the authors used only one dataset and one preprocessing-based defense.
    - First, the authors should evaluate the method over datasets other than CIFAR-10. I suggest datasets with subtle feature differences between classes. CIFAR-10 includes images of objects whose difference is apparent, so adding Gaussian noise will not damage semantic information. For example, adding noise to a frog image will not make it look like an airplane. Therefore, the Gaussian noise in each diffusion step influences the final classification less.
However, some datasets include classes with subtle differences and can damage the semantic information in the image. GTSRB (German Traffic Sign Recognition Benchmark) contains photos of traffic signs. Some traffic signs have similar shapes with different figures inside, e.g., all left turn, right turn, and go straight are all “white figures in a blue circle.” If noise damages these white figures, we lose the semantic information to classify correctly at the end. Also, images of numbers, such as SVHN (Street View House Numbers), contain visually similar classes, e.g., 8 and 9. Because the defense keeps adding noise to the input, the authors should evaluate their defense over such datasets to ensure that the method will not damage the semantic feature more than needed.
   - Also, the authors should provide the performance of existing preprocessing-based defenses other than Yoon et al. Even though this is the leading method in preprocessing-based defenses, an experimental study should be more comprehensive than a single method on a single dataset.
    - Last, while the authors provided several attack methods, the author should evaluate their defense against more attacks. This paper should emphasize the “threat model-agnostic defense” because it is a rare virtue in this field. To this end, the variation in the attack methods is not enough. Consider more experiments against different attacks: at least against FGSM and CW.
    - In general, comparing methods on different architecture is not a fair comparison. Because different networks will have different baseline accuracy (the accuracy before the adversarial attack), and the bad performance may be because of a bad choice of architecture. The best fix is to adversarially train a single architecture with all the adversarial training methods. If the authors cannot take this option due to the time constraint, at least the baseline accuracy should be presented. Even with the baseline accuracies, the authors should provide the performance of their methods on all the used architecture to make a fair comparison.
3. Presenting the images’ change over the process in the Appendix might be a good idea. This change of images will show that the added Gaussian noise does not harm the semantic information too much and will show the denoising process visually. Also, by visually displaying the intermediate output, the authors can demonstrate how a deep diffusion depth changes classification during the procedure.

**Strengths And Weaknesses:**

**[[Strength]]**

1. The proposed idea sounds like a practical approach that effectively adds randomness in the preprocessing step.
2. The authors evaluated the proposed defense against various attack scenarios, including BPDA and EOT.
3. Looking at the current evaluation, the performance of the proposed defense seems promising. The defense rate against EOT is remarkable, considering EOT is an effective attack method against a randomized defense.

**[[Weaknesses]]**
1. The paper does not contain enough information to reproduce the result. The authors should describe the attack scenarios in detail. For example, the author mentioned that stochastic preprocessing is challenging for white-box attacks. The suggested defense uses stochastic preprocessing, so what are a grey-box attack and a white-box attack against this defense?
2. Generally, more comprehensive experiments are required to convince the readers. The authors tested several attack methods for evaluation; however, they needed to evaluate more different datasets and preprocessing-based defenses.
3. While the authors used various attacks against different adversarial training methods, those methods use different architectures. In this case, the result doesn’t seem fair. First, the low performance of adversarial training might be due to a bad baseline accuracy. Second, the authors did not demonstrate their method on different model architectures.

---

### Author Response · Authors · 2023-01-01
**Reply**

Thank you for your constructive feedback. We were not familiar with the work done by [1] and it is indeed similar to our work. However, the evaluation is different since [1] uses an estimation for the gradients while we are using the true values. We will keep researching our method and will shed the light on our contribution.

[1] Nie, W., Guo, B., Huang, Y., Xiao, C., Vahdat, A., & Anandkumar, A. (2022). Diffusion Models for Adversarial Purification. ICML 2022.

---

### Note · Authors · 2023-01-03

I have read and agree with the venue's withdrawal policy on behalf of myself and my co-authors.